# Non-stationary Experimental Design under Linear Trends

**David Simchi-Levi** [‡§], **Chonghuan Wang** [§], **Zeyu Zheng** [†]

[‡] Institute for Data, Systems, and Society, Department of Civil and Environmental Engineering,
Operations Research Center, MIT

[§] Laboratory for Information and Decision Systems, MIT

[†] Department of Industrial Engineering and Operations Research, University of California, Berkeley

`dslevi@mit.edu, chwang9@mit.edu, zyzheng@berkeley.edu`

## Abstract

Experimentation has been critical and increasingly popular across various domains, such as clinical trials and online platforms, due to its widely recognized benefits. One of the primary objectives of classical experiments is to estimate the average treatment effect (ATE) to inform future decision-making. However, in healthcare and many other settings, treatment effects may be non-stationary, meaning that they can change over time, rendering the traditional experimental design inadequate and the classical static ATE uninformative. In this work, we address the problem of non-stationary experimental design under linear trends by considering two objectives: estimating the dynamic treatment effect and minimizing welfare loss within the experiment. We propose an efficient design that can be customized for optimal estimation error rate, optimal regret rate, or the Pareto optimal trade-off between the two objectives. We establish information-theoretical lower bounds that highlight the inherent challenge in estimating dynamic treatment effects and minimizing welfare loss, and also statistically reveal the fundamental trade-off between them.

## 1 Introduction

Experimental design plays a critical role in conducting research across various fields including clinical trials (see, e.g., [75], [14], [69]) and online platforms (see, e.g., [52], [1], [44], [21]). It enables the evaluation of the effectiveness of treatments, interventions, or policies in a rigorous and systematic manner, protecting the safety and well-being of participants and the wider population. Typically, one of the primary objectives of conducting experiments is to estimate the average treatment effect (ATE), which represents the difference in outcomes between different treatments or controls. The outcomes of each treatment are assumed to be stationary by default (see, e.g., [48]), ensuring that the information collected and conclusions drawn from in-experiment periods are informative for future decision-making. However, in some situations, ATE may be non-stationary, meaning that it evolves over time. Here, we present one illustrative example from clinical trials to highlight the importance of considering non-stationarity in ATE estimation.

Malaria has been one of the world's deadliest diseases, with approximately 247 million cases and 618 thousand deaths reported in 2021 according to the World Health Organization (WHO). Antimalarial drug resistance in Africa has been highlighted by the world malaria report (see, [86]) as one the global key events of malaria control and elimination efforts in 2021-2022. *P. falciparum*, the deadliest species of *Plasmodium* that causes malaria in humans, has shown resistance to first-line treatments such as artemether-lumefantrine (AL), artesunate-amodiaquine (AS-AQ), artesunate-pyronaridine (AS-PY), and dihydroartemisinin-piperaquine (DHA-PPQ) in Africa recently. Alarmingly, at least four recent studies have reported treatment failure rates greater than 10% after treatment with AL

in Burkina Faso (see, [35]) and Uganda (see, [91]). Two study sites in Burkina Faso also reported treatment failure rates greater than 10% after treatment with DHA-PPQ. These alarming findings underscore the importance of understanding and estimating how the treatment effects evolve over time, i.e., how rapidly drug resistance escalates. Failure to account for non-stationarity in treatment effect may lead to suboptimal treatment plans, harming social welfare.

Given the ubiquity of non-stationarity, ignoring such a structure can lead to misleading information and poor decision-making. Figure 1 provides an illustrative example where non-stationarity can be well captured by linear models. If we use traditional randomized control trials and classical estimators based on sample averages without considering non-stationarity, we may conclude that the treatment is always better than the control. However, the linear non-stationarity implies that we need to pay more attention since the treatment outcomes decrease much faster than the control. This means that the treatment does not always outperform the control during the after-experimental periods. In the context of the drug resistance example, this phenomenon can be interpreted as follows: a new treatment may have a significant positive effect during the experiment, but drug resistance develops much faster than the control. Therefore, we cannot always rely on the treatment after the experiment. Understanding how treatment effects evolve over time is thus crucial for making informed decisions.

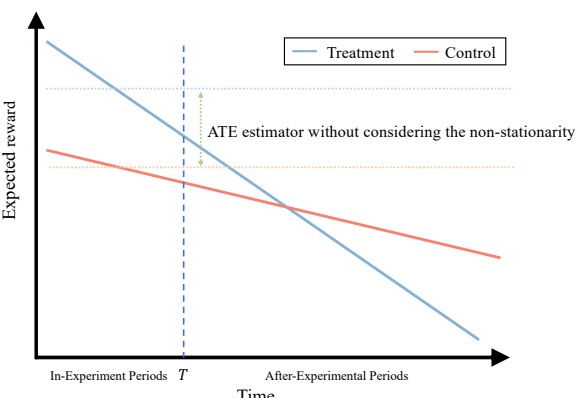

Figure 1: An illustrating example of the non-stationarity.

In this paper, we focus on the challenge of designing experiments in non-stationary environments. To tackle this problem, we first need to answer the question of *how to formulate non-stationarity*. If there is no underlying structure to the non-stationarity, such as when outcomes are determined by a malicious adversary, then it is not justifiable to conduct experiments because there is little information that we can infer for future decision-making. In this work, we limit our attention to scenarios where potential outcomes follow structured trends. Specifically, we consider situations where outcomes are linear with time, which provides a suitable starting point for research on experimental design under non-stationarity. Such structures also help us to determine meaningful metrics to use instead of the traditional static ATE, since it is no longer informative enough in non-stationary environments. While other ways to formulate non-stationarity can be found in the literature (e.g., [15], [16], [24], [10], [36]), we carefully argue in Section 1.2 that these traditional formulations may not suffice for our purposes.

In addition to metrics related to the treatment effect, non-stationarity can also increase attention to the welfare loss issue during experiments. Specifically, every time a suboptimal treatment is assigned, there is a welfare loss as the unit could have been treated better with the optimal treatment. In a stationary environment, this welfare loss remains constant over time. However, non-stationarity can lead to an increasing welfare loss over time. Thus, it might not be ideal to ignore the welfare loss during the experiment, particularly when the in-experiment periods are not negligible compared to the after-experiment periods. In this work, we also aim to address *how to control welfare loss during experiments* and *explore the relationship between minimizing welfare loss and maximizing the statistical power of estimating the dynamic treatment effect*.

## 1.1 Main results and contribution

The primary contribution of this work is the presentation of a framework that enables experimenters and researchers to design reliable, efficient, and flexible experiments in structured non-stationary environments. To the best of our knowledge, this is the first study of the dynamic treatment effect in the experimental design literature, extending the classical static ATE. We believe that this work could serve as a starting point for further research on designing experiments in non-stationary environments.

We examine what is arguably the most foundational scenario, where the expected potential outcomes of each treatment are represented by linear functions of time $t$. In this context, the dynamic treatment

effect can be described by a linear function as well. We introduce a flexible experimental design framework that draws upon the well-established principle of Optimism in the Face of Uncertainty from the online learning literature [29][2]. Our design can be readily customized to suit various notions of optimality. Specifically, in this work, we systematically explore optimality from three distinct perspectives: optimal estimation error rate, optimal regret rate, and Pareto optimality.

**The optimal estimation error rate design** maximizes statistical power for inferring dynamic treatment effects. We establish a lower bound of $\frac{1}{\sqrt{T}}$ for the error rate of any estimator, where $T$ is the total experimental horizon length. Our design achieves this lower bound by selecting appropriate input parameters. **The optimal regret rate design** minimizes in-experiment welfare loss (regret) in the worst case scenario. Traditional regret analysis techniques are not applicable due to unbounded outcomes. Our design achieves an $\sqrt{T}$ upper bound for regret rate by selecting input parameters, matching the worst-case lower bound. Linear non-stationarity in our problem may result in a regret rate of $T^2$ without careful design, unlike the stationary setting where regret is bounded by $T$. **The Pareto optimal design** strikes an optimal and flexible balance between the two objectives of maximizing statistical power and minimizing regret. We reveal the inherent and fundamental trade-off between these objectives statistically. Specifically, among all designs that can ensure a regret upper bound of $T^\beta$, the smallest error any estimator can achieve is $T^{-\frac{\beta}{4}}$ in the worst case. Conversely, among all designs that can guarantee an error bound of $T^{-\frac{\beta}{4}}$, a regret of order $T^\beta$ is unavoidable. Our design achieves different levels of trade-off by flexibly selecting input parameters.

## 1.2 Literature Review

**Inference under non-stationarity.** Considerable literature focuses on inference and decision-making under non-stationarity. One area is the inference for non-stationary Poisson processes [4][63][53] [96][39]. [39] is relevant as they establish statistical inference models for non-stationary arrival models with linear trends. [88] shows that ignoring non-stationarity can lead to inefficient or invalid A/B tests. They propose a new framework for statistical analysis of non-stationary A/B tests.

**Decision-making under non-stationarity.** This work relates to adaptive decision-making in non-stationary environments. Existing literature has two main streams. The first stream focuses on budget-based drifting environments, where changes are dynamically and adversarially made within a variation budget [15][16][17][47][84][61][24][80]. The second stream considers piecewise stationary/switching environments, with adaptive assignment policies designed for various bandit settings [10][36][58][22][47][61][89][18]. However, these formulations lack the knowledge transferability to after-experimental periods, which is essential for our purpose. Recent studies address bandit experiments with non-stationary contexts [70], provide information-theoretic analysis [65], propose improved algorithms [59], and define non-stationary bandits more rigorously [60]. Moreover, various bandit models consider specific non-stationary reward structures, such as recharging bandits [51][90][66], restless bandits [85][81][94], blocking bandits [13][12][20][8], rotting bandits [56][71][72], and rising bandits [64][28]. In operations, assortment decisions under non-stationarity are studied [34], as well as pricing with evolving demands [50][42][99]. [89] study the recommendation problem. Recent work extends bandit analysis to reinforcement learning [62][97][25]. Some works focus on optimizing with full feedback [26][16][43].

**Adaptive Experimental Design.** Multi-armed bandit is an efficient paradigm for adaptive experiments [77][54][9]. [78] and [3] apply diffusion-asymptotic analysis to study randomized experiments, including bandit problems. [48] consider experimental design for choosing the best arm with arriving units in waves. [30] integrate adaptively weighted doubly robust estimator into Thompson sampling. [32] incorporate synthetic control into MAB for interference scenarios. [73] also investigate the trade-off between regret and causal inference using Pareto optimality. Non-stationarity that we consider here greatly impacts Pareto optimality structure and reveals stochastic bandit as a special case. [41] design a two-stage experiment for ATE estimation. [6] model adaptive assignment as a Markov decision process, highlighting suboptimality of RCTs. [49] propose sequential testing framework with finite and infinite sample analysis. [87] consider an optimization problem in adaptive experiments. [38] transform experimental design with temporal interference into a Markov decision problem. [19] optimize subject allocation for ATE estimation precision. [95] address adaptive experimental design for sequential unit assignment with heterogeneous covariates.

**Post-experiment Inference.** There is a substantial literature on post-experiment inference from logged adaptively collected data where adaptive experiment data is a typical example. One of

the central tasks along this line is the evaluation of a new policy given historic data (see, e.g., [31][76][57][83][46][7][92][98]). [23] provide a procedure to adopt bootstrap on the data collected by a bandit to debias the sample means. [40] construct confidence intervals for policy evaluation, including ATE, in adaptive experiments. Works along this line usually impose some "overlapping" assumptions on the data collection process. Since we are jointly considering the experimentation and the inference, such an overlapping assumption may harm the welfare gained within the experiment.

Finally, we remark that the full version of this paper (containing additional theoretical results, extensions, and missing proofs) is available at `https://ssrn.com/abstract=4514568`.

## 2 Experimental Design with Linear Trends

### 2.1 Formulation

We establish the formulation of our non-stationary experiments with a finite set of arms $\mathcal{A}$. Without loss of generality, we focus on $\mathcal{A} = \{1, 2\}$ representing a treatment and a control. The total length of the experiment is denoted as $T$. At each time $t \in [T]$[1], the environment generates potential outcomes $r_t(a)$ for each arm $a \in \mathcal{A}$. The chosen arm $a_t$ determines the observed outcome $r_t = r_t(a_t)$. Our study focuses on non-stationary experiments with linear trends, specifically considering the follows,

$$r_t(a) = \theta_{a,0} + \theta_{a,1}t + \varepsilon_{a,t} := \theta_a^\top \phi(t) + \varepsilon_{a,t}, \tag{1}$$

where $\phi(t) = (1, t)^\top$, $\theta_a \in \mathbb{R}^2$ is fixed throughout the experimental horizon $T$ but unknown to the decision maker. For a specific arm $a$, $\{\varepsilon_{a,t}\}_{t \geq 1}$ are independent and identically distributed (i.i.d.) mean-zero $\sigma_a^2$-sub-Gaussian random variables (r.v.'s), i.e., $\mathbb{E}[e^{\lambda \varepsilon_{a,t}}] \leq \exp(\lambda^2 \sigma_a^2 / 2)$ for any $\lambda \in \mathbb{R}$. For simplicity, we also assume for $\sigma_a \leq \sigma_0 < \infty$ for all $a$. Note that while $\theta_a$ remains fixed, the distribution of the reward of arm $a$, $r_t(a)$, evolves over time, thereby satisfying the definition of non-stationarity as outlined in [60]. Another mild assumption we make is the boundedness of $\theta_a$, i.e., $\|\theta_a\| \leq S$ for all $a \in \mathcal{A}$, which is commonly used (see, e.g., [27], [2], and [55]). A non-stationary experiment instance with linear trends can be denoted by $\nu = (P_1, P_2)$, where $P_i$ is the distribution of the rewards of arm $i$. The optimal arm at time $t$ is the one with the maximum expected reward, which is denoted by $a_t^* := \arg\max_{a \in \mathcal{A}} \theta_{a,0} + \theta_{a,1}t$, and thus is also unknown and may even change over time. The dynamic treatment effect can be fully captured by the column vector $\theta_\nu := \theta_1 - \theta_2 \in \mathbb{R}^2$. Denote all non-stationary experiment instances with linear trends to constitute a feasible set $\mathcal{E}_1$.

At every time $t$, the decision maker observes the history $\mathcal{H}_{t-1} = (a_1, r_1, \cdots, a_{t-1}, r_{t-1})$. An *admissible* design $\pi = \{\pi_t\}_{t \geq 1}$ maps the history $\mathcal{H}_{t-1}$ to an action $a_t$. We use the traditional accumulative *regret* to measure the efficiency of online learning, defined as the difference between the expected reward under the clairvoyant optimal policy and the policy $\pi$, i.e., $\mathcal{R}_\nu^\pi(T) = \sum_{t=1}^T \theta_{a_t^*}\phi(t)^\top - \theta_{a_t}\phi(t)^\top$. Note that $\mathcal{R}_\nu^\pi(T)$ is still a random variable because $\{a_t\}_{t \geq 1}$ are still random. Similarly, an admissible estimator $\hat{\theta}_t$ of the treatment effect at time $t$ maps $\mathcal{H}_t$ to an estimate of $\theta_\nu$. The quality of the estimator can be measured by the norm of the difference between $\hat{\theta}_t$ and $\theta_\nu$, for example, the $l_1$-norm $\|\hat{\theta}_t - \theta_\nu\|_1$. *The design of experiment includes designing $\pi$ (i.e., how to make an adaptive assignment decision) and $\hat{\theta}_T$ (i.e., how to conduct causal inference).*

**Remark 1:** Linear trends, admittedly, is a strong assumption, but they have proven to be highly useful for understanding and predicting certain phenomena across various fields. [82] observe a linear trend in the prevalence of obesity and overweight among US adults, based on national survey data collected between 1970s and 2004. This linear trend served as a benchmark for predicting the prevalence from 2004 to 2050, influencing numerous epidemiology studies. [37] conduct a study on monitoring and predicting the California sea otter population using linear trends observed in aerial surveys, showcasing the usefulness of linear trends in ecological studies. [68] reveal a linear trend in the integrated water vapor content based on a 10-year ground-based GPS data, highlighting the application of linear trends in atmospheric research

**Remark 2:** The traditional multi-armed bandit problems are primarily focused on designing $\pi$ to have minimal regrets (see, e.g., [55]). However, in our design, we have an equally important task of designing $\hat{\theta}_T$. Furthermore, we outline the unique challenges and opportunities we face compared to the stochastic bandits and contextual bandits studied in the current literature.

---

[1]Throughout this paper, we define $[n] := \{1, \cdots, n\}$, $a \vee b := \max\{a, b\}$ for $a, b \in \mathbb{R}$ and $\|u\|_V := \sqrt{u^\top V u}$ for the positive definite matrix $V$.

Our problem encompasses the well-studied stationary stochastic multi-armed bandit problem (e.g., [54], [9]) by setting $\theta_{a,1} = 0$ in Eq. (1) for all $a \in \mathcal{A}$. The non-stationary structure introduces new challenges, such as the possibility of the optimal action varying over time, requiring additional efforts to control the regret. Second, while $(1, t)^\top$ in Eq. (1) can be seen as a special kind of context connecting our problem to the contextual bandit problem (e.g., [27]), the first difference lies in that the context $(1, t)^\top$ is fixed and known at each time period. However, a challenge arises as we can no longer assume the context $(1, t)^\top$ to be bounded, as commonly done in the literature. Some works assume the $l_2$-norm to be no larger than a universal constant, say 1 (see, e.g., [27] and [24]). A straightforward solution to address this issue may involve relaxing the bound of the context to $\sqrt{T^2 + 1}$ if the experimental horizon $T$ is known in advance. However, such a relaxation might be too loose, particularly when analyzing regret upper bounds. A more thorough discussion on this relaxation is deferred to the next subsection when we analyze the regret of our new design.

## 2.2  Design of Experiments with Linear Trends

We present our design called *Linear Exploration and OFU* (L-EOFU) design in Algorithm 1. The term OFU refers to the well-known principle of Optimism in the Face of Uncertainty, which arises from and has been widely adopted in the multi-armed bandit literature (see, e.g., [29], [2] and their follow-ups). For brevity, we use the notation L-EOFU($\alpha$) to emphasize the crucial input parameter $\alpha$, whose role will become evident in the subsequent discussion. Our design operates in a two-phase manner: the exploration phase and the OFU phase. We now elaborate on each phase in detail.

The exploration phase is the first steps of our design, which lasts for approximately $T^\alpha$ periods. The parameter $\alpha$ plays a crucial role in controlling the length of this phase. During this phase, our L-EOFU algorithm deterministically alternates between playing arms 1 and 2. In Algorithm 1, we choose to play arm 1 when the time index $t$ is odd and arm 2 when $t$ is even. This deterministic approach preserves the independence between the collected data. By the end of this phase, the decision-maker obtains approximately $\frac{T^\alpha}{2}$ independent samples for each arm, although these samples may not be identically distributed. L-EOFU will utilize the data collected during this initial phase to produce an estimator for the treatment effect, denoted as $\hat{\theta}_T^{\text{L-EOFU}(\alpha)}$ in line 18 of the algorithm. Specifically, we estimate $\theta_1$ and $\theta_2$ separately by regressing the rewards against the time $t$ using the data collected up to time $T^\alpha$, i.e., $\hat{\theta}_{1,T}^{\text{L-EOFU}(\alpha)} = (\sum_{t \in [[T^\alpha]]} \phi(t)\phi(t)^\top)^{-1} \sum_{t \in [[T^\alpha]]} r_t \phi(t)$, where $[[T^\alpha]]$ denotes the set of all the odd numbers that are no larger than $T^\alpha$. The final estimate of $\theta_\nu$ is $\hat{\theta}_T^{\text{L-EOFU}(\alpha)} = \hat{\theta}_{1,T}^{\text{L-EOFU}(\alpha)} - \hat{\theta}_{2,T}^{\text{L-EOFU}(\alpha)}$. While the estimator we use may be sample-inefficient as we do not utilize the $T - T^\alpha$ samples in the second phase, we will show in the next subsection that the data collected in the first stage is sufficient to achieve a rate-optimal estimator of the treatment effect. Intuitively, if one arm consistently outperforms the other, most of the data in the second phase comes from the optimal arm, with fewer observations from the suboptimal arm. Thus, observations from the suboptimal arm are mostly obtained during the first stage. Moreover, the bottleneck for estimating the treatment effect is very likely determined by the arm with the fewest observations, which is typically the suboptimal arm. Although we discard much data from the optimal arm when conducting inference, the bottleneck is still the suboptimal arm whose most information will be collected through the first phase. Additionally, we choose not to use all the data since adaptively collected data in the second stage can harm the independence of the data, leading to undesirable statistical properties such as bias. This issue has been pointed out in a large body of recent work [67][93][23]. Formally, we have the following Theorem 2 characterizing the properties of the estimator we choose.

**Theorem 1** *Assuming $T^\alpha \geq 4$, for any $\nu \in \mathcal{E}_1$, there exists a constant $c_1 > 0$ such that for any $\epsilon > 0$, with probability at least $1 - \epsilon$, $\left\| \hat{\theta}_T^{\text{L-EOFU}(\alpha)} - \theta_\nu \right\|_1 \leq c_1 T^{-\frac{\alpha}{2}} \sqrt{\log \frac{8}{\epsilon}}$, where $c_1 = \frac{128}{3}(\sigma_1 \vee \sigma_2)$. Moreover, we have $\max_{\nu \in \mathcal{E}_1} \mathbb{E}\left[ \left\| \hat{\theta}_T^{\text{L-EOFU}(\alpha)} - \theta_\nu \right\|_1 \right] = \mathcal{O}(T^{-\frac{\alpha}{2}})$.*

Our assumption that $T^\alpha \geq 4$ aligns with our intuition that a minimum of two samples per arm is required to identify a linear structure. Theorem 1 demonstrates that the bound $T^{-\frac{\alpha}{2}}$ decreases as $\alpha$ increases, indicating that a longer exploration period leads to better estimation of the dynamic treatment effect. When $\alpha$ is set to 1, we conduct pure exploration throughout the entire experimental horizon, similar to the well-known switchback experiments [21]. In this case, the estimation error can be controlled to the order of $\mathcal{O}(1/\sqrt{T})$, which aligns with traditional statistical results of linear regression [79]. Moreover, our analysis can be extended to any $l_p$-norm, with only slight variations

**Algorithm 1:** Linear Exploration and OFU (L-EOFU)

1 **Input:** Controlling parameter $\alpha$, instance parameters: $\sigma_a$ for all $a \in \{1, 2\}$, $T, S, \delta$
2 **Initialization:** $\lambda = 1$, $\mathcal{D}_{1,0} = \mathcal{D}_{2,0} = \mathcal{D}_{1,T} = \mathcal{D}_{1,T} = \emptyset$, $\phi(\cdot) = (1, \cdot)^\top$
3 **for** $t = 1, 2, \cdots, \lceil T^\alpha \rceil$ **do** // Phase 1: Exploration
4     Select $A_t = 1$ if $t$ is odd; otherwise select $A_t = 2$;
5     Observe reward $r_t$;
6     $\mathcal{D}_{A_t,0} \leftarrow \mathcal{D}_{A_t,0} \cup \{(\phi(t), r_t)\}$ and $\mathcal{D}_{A_t,T} \leftarrow \mathcal{D}_{A_t,T} \cup \{(\phi(t), r_t)\}$;
7 **end for**
8 **for** $t = \lceil T^\alpha \rceil + 1, \lceil T^\alpha \rceil + 2, \cdots, T$ **do** // Phase 2: OFU
9     $V_{a,t} = \sum_{(\phi(i),r_i) \in \mathcal{D}_{a,T}} \phi(i)\phi(i)^\top + \lambda I$ for $a \in \{1, 2\}$;
10     $\gamma_{a,t} = \sigma_a \sqrt{2 \log(\frac{2 \det(V_{a,t})^{1/2} \det(\lambda I)^{-1/2}}{\delta})} + \lambda^{1/2} S$ for $a \in \{1, 2\}$;
11     $\tilde{\theta}_{a,t} = V_{a,t}^{-1} \sum_{(\phi(i),r_i) \in \mathcal{D}_{a,T}} r_i \phi(i)$ for $a \in \{1, 2\}$;
12     $\tilde{R}_t(a) = \tilde{\theta}_{a,t}^\top \phi(t) + \gamma_{a,t} \|\phi(t)\|_{V_{a,t}^{-1}}$ for $a \in \{1, 2\}$;
13     Select $A_t = \arg\max_{a \in \{1,2\}} \tilde{R}_t(a)$;
14     Observe reward $r_t$;
15     $\mathcal{D}_{A_t,T} \leftarrow \mathcal{D}_{A_t,T} \cup \{(\phi(t), r_t)\}$;
16 **end for**
17 $\hat{\theta}_{a,T}^{\text{L-EOFU}(\alpha)} = (\sum_{(\phi(i),r_i) \in \mathcal{D}_{a,0}} \phi(i)\phi(i)^\top)^{-1} \sum_{(\phi(i),r_i) \in \mathcal{D}_{a,0}} r_i \phi(i)$ for $a \in \{1, 2\}$;
18 **Output:** $\hat{\theta}_T^{\text{L-EOFU}(\alpha)} = \hat{\theta}_{1,T}^{\text{L-EOFU}(\alpha)} - \hat{\theta}_{2,T}^{\text{L-EOFU}(\alpha)}$;

in the constant $c_1$. Additionally, note that the high probability bound can be used for constructing *confidence intervals* and conduct *hypothesis testing*.

The second phase of our L-EOFU follows the OFU principle. Specifically, at each time period $t$, L-EOFU calculates an optimistic estimate of the rewards for each arm, denoted as $\tilde{R}_t(a)$ in line 12. This estimate is referred to as optimistic because, with probability at least $1 - \delta$, for all $t \in [T]$ and $a \in \{1, 2\}$, $\tilde{R}_t(a) = \tilde{\theta}_{a,t}^\top \phi(t) + \gamma_{a,t}\|\phi(t)\|_{V_{a,t}^{-1}} \geq \theta_a^\top \phi(t) = \mathbb{E}[R_t(a)]$, where $\gamma_{a,t} \leq \sigma_a(\sqrt{12 \log(T) - 2 \log(4.5\lambda\delta)} + \lambda^{1/2} S)$ and by default, $\lambda$ is set to be 1. The proof of this claim follows the standard techniques outlined in Theorem 2 of [2]. Based on the optimistic estimation, L-EOFU selects the arm with the highest optimistic estimated value. The OFU principle has been shown to strike a delicate bal-

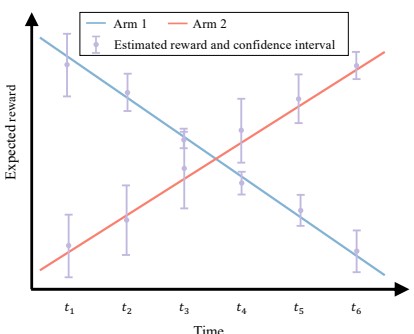

Figure 2: An illustrative example on OFU.

ance between exploration and exploitation in the bandit literature [2][55]. As mentioned earlier, one of the challenges in our problem is the possibility of the optimal arm changing over time. The OFU principle has an additional advantage in that it can automatically and implicitly detect when the optimal arm switches from one to another. To illustrate this, Figure 2 presents an example of how OFU behaves when the optimal arm switches. Initially (before $t_3$), arm 1 outperforms arm 2. According to L-EOFU, we continue to play arm 1, resulting in a relatively fast shrinkage of the confidence interval for the estimated values of arm 1. Conversely, for arm 2, the confidence interval even expands as $\phi(t)$ increases in magnitude. As time approaches $t_3$, the difference between arm 1 and arm 2 gradually diminishes. At time period $t_3$, the optimistic estimation of arm 2 surpasses that of arm 1, prompting L-EOFU to start playing arm 2. This addresses the challenge mentioned earlier. The following theorem presents the regret upper bound.

**Theorem 2** *For any $\nu \in \mathcal{E}_1$ and $0 < \alpha \leq 1$, with probability at least $1 - \delta$,*

$$\mathcal{R}_\nu^{L\text{-}EOFU(\alpha)}(T) \leq 4ST^{2\alpha} + 4(\sigma_1 \vee \sigma_2)(\sqrt{12 \log(T) - 2 \log(4.5\lambda\delta)} + \lambda^{1/2}S)\sqrt{\frac{4^{1+\frac{2}{\alpha}}}{\ln 2} \ln(2T)T}, \quad (2)$$

*where $\lambda$ is initialized as 1. Furthermore, $\delta = 1/T$ leads to $\mathbb{E}[\mathcal{R}_\nu^{L\text{-}EOFU(\alpha)}(T)] = \tilde{\mathcal{O}}(T^{2\alpha} \vee \sqrt{T})$.*

Equation (2) can be decomposed into two terms: a $T^{2\alpha}$ term and a $\sqrt{T \ln(T)}$ term. The first term arises from the exploration stage, as $\sum_{t=1}^{T^\alpha} t = \mathcal{O}(T^{2\alpha})$ intuitively. The second term is a result of the

OFU phase. Only when $\alpha \geq \frac{1}{4}$ does the first term dominate the second term. In fact, there is little motivation for a decision maker to choose $\alpha < \frac{1}{4}$, as we will discuss in detail in the next subsection. For pure theoretical interest, the regret behavior in the region around $\alpha = 0$ could be studied by choosing $\alpha = 1/\ln\ln(T)$, given that there is an $\alpha$ in the denominator. For those familiar with the bandit literature, a similar $\mathcal{O}(\sqrt{T \ln(T)})$ bound can be obtained in the stochastic contextual bandit setting [2]. As we mentioned before, one possible approach to reduce our problem to the one in [2] is by treating $\phi(t)$ as a special kind of context. However, a key difference is that the context in [2] is assumed to be bounded by a constant, whereas in our setting, the bound should be $\sqrt{1 + T^2}$ if we know $T$ in advance. Consequently, to apply all the techniques from [2], $\lambda$ in Equation (2) should be chosen as $\Theta(T)$ instead of 1. However, such a choice of $\lambda$ would yield a regret bound of $\mathcal{O}(T)$, which is significantly weaker than the $\mathcal{O}(\sqrt{T})$ regret we derived. Furthermore, in the stochastic contextual bandit setting, where everything is assumed to be bounded, the regret can always be upper bounded by $\mathcal{O}(T)$ for any policy, and thus only sublinear regrets are meaningful. However, here, with the linear trend, the largest possible regret for a policy without careful design can be on the order of $\mathcal{O}(T^2)$. Thus, the $\mathcal{O}(T^{2\alpha})$ regret bound is still informative even if $1/2 < \alpha < 1$.

Another important point to emphasize is that although the regret can be decomposed into two terms, it does not mean that we can analyze each phase independently of the other. The $T^\alpha$ samples we collect from the first stage play a crucial role in our analysis for the second phase. This is evidenced by the fact that $\alpha$ not only appears in the first term, but also takes effect in the second term by a non-trivial way. Specifically, the regret of the second phase can be upper bounded by the sum of the length of the confidence interval at each time, i.e., the regret at time $t$ can be bounded by $2\gamma \|\phi(t)\|_{V_{1,t}^{-1}} \mathbb{I}_{A_t=1} + 2\gamma \|\phi(t)\|_{V_{2,t}^{-1}} \mathbb{I}_{A_t=2}$, where $V_{a,t}$ is defined to be $\sum_{i=1}^{t} \phi(i)\phi(i)^\top \mathbb{I}_{A_i=a} + \lambda I$. The $T^\alpha$ samples collected in the first stage are efficient in controlling the minimum eigenvalue of $V_{a,t}$, and thus be useful in controlling the magnitude of the regret.

In addition, we would like to compare our two-stage design with the traditional exploration-then-exploitation design [55]. The similarity between the two lies in the fact that the first exploration phase provides valuable information for future decision-making, as we discussed earlier. However, the key difference is that exploration-then-exploitation typically only exploits the information gathered during the first exploration phase and does not continue to learn new information during the second phase. In contrast, our `L-EOFU` algorithm continues to and has to learn throughout the entire learning period, since the environment may rapidly evolve over time. Moreover, the exploration phase in `L-EOFU` also plays an important role in conducting causal inference. Therefore, when designing the length of the first phase, we must consider not only the regret but also the quality of causal inference. One reason why we move away from the exploration-then-exploitation method is that it may not always guarantee the optimal regret rate. Another important motivation for conducting pure exploration first is that the cost of exploration may increase linearly with time.

## 2.3   Optimality of The Design

In this subsection, we will examine the optimality of our `L-EOFU` design from three different perspectives. Firstly, we will consider two extremes where the decision maker is solely concerned about either the estimation error of the dynamic treatment effect or the regret. Subsequently, when jointly considering the two objectives, we will introduce the concept of Pareto optimality, which describes a scenario where neither the statistical power of estimating the treatment effect nor the regret can be improved without compromising the other. First, we focus on the best achievable quality of the inference of $\theta_\nu$, regardless of the regret incurred during the experiment. The following lemma describes the intrinsic difficulty of conducting inference for the dynamic treatment effect.

**Lemma 1** *For any constant $c_2 > 0$, $\inf_{\hat{\theta}} \sup_{\nu \in \mathcal{E}_1} \mathbb{P}_\nu \left( \left\| \hat{\theta} - \theta_\nu \right\|_1 > \frac{2\sqrt{2}c_2\sigma_0}{\sqrt{T}} \right) \geq \frac{1}{2} - c_2$. Moreover, $\inf_{\hat{\theta}} \sup_{\nu \in \mathcal{E}_1} \mathbb{E}_\nu \left[ \left\| \hat{\theta} - \theta_\nu \right\|_1 \right] \geq \frac{\sqrt{2}\sigma_0}{8\sqrt{T}}$.*

This lemma establishes an information-theoretic lower bound on the best achievable quality of inference for $\theta_\nu$. It indicates that, for any estimator $\hat{\theta}$, there will always exist a challenging instance that results in the estimator being at least $\Theta(1/\sqrt{T})$ away from the true value. Our proof idea is to reduce the two-dimensional estimation problem to a one-dimensional problem. Specifically, we consider a subset of all the problem instances $\mathcal{E}_0 := \{\nu \in \mathcal{E}_1 : \theta_{a,1} = 0, \forall a \in \{1, 2\}\}$, which includes all the stationary instances (i.e., without time trends), and this provides a natural lower

bound that $\inf_{\hat{\theta}} \sup_{\nu \in \mathcal{E}_1} \mathbb{P}_\nu \left( \left\| \hat{\theta} - \theta_\nu \right\|_1 > \frac{2\sqrt{2}c_2\sigma_0}{\sqrt{T}} \right) \geq \inf_{\hat{\theta}} \sup_{\nu \in \mathcal{E}_0} \mathbb{P}_\nu \left( \left\| \hat{\theta} - \theta_\nu \right\|_1 > \frac{2\sqrt{2}c_2\sigma_0}{\sqrt{T}} \right)$.
For the instance class $\mathcal{E}_0$, the problem is reduced to estimating only the intercept terms. By choosing $\alpha = 1$, Theorem 1 indicates $\|\hat{\theta}_T^{\text{L-EOFU}(1)} - \theta_\nu\|_1$ can be upper bounded by $\mathcal{O}(1/\sqrt{T})$, which matches with the lower bound we derive in Lemma 1 in terms of the dependence of the order of $T$. Therefore, we can state the following theorem.

**Theorem 3 (Optimal Error Rate Design)** *$\hat{\theta}_T^{\text{L-EOFU}(1)}$ achieves the optimal $1/\sqrt{T}$ error rate, which can not be improved up to a constant factor. Moreover, $\inf_{\hat{\theta}} \sup_{\nu \in \mathcal{E}_1} \mathbb{E}_\nu \left[ \left\| \hat{\theta} - \theta_\nu \right\|_1 \right] = \Theta(1/\sqrt{T})$.*

Theorem 3 further establishes that the existence of $\hat{\theta}_T^{\text{L-EOFU}(1)}$ confirms the tightness of the $1/\sqrt{T}$ lower bound derived in Lemma 1. As mentioned earlier, setting $\alpha = 1$ corresponds to the simple switchback experimental design. Theorem 3 implies that even in the presence of linear trends, switchback experiments can still maintain strong statistical power. However, there is no free lunch. Choosing $\alpha = 1$ results in a regret upper bound of $\mathcal{O}(T^2)$, which can be highly undesirable if minimizing regret is also one of the objectives. Now, we turn to discuss about the other extreme that the regret minimization is the only objective. We present the minimax regret lower bound as follows.

**Lemma 2 (Regret Lower Bound)** $\inf_\pi \sup_{\nu \in \mathcal{E}_1} \mathbb{E} \left[ \mathcal{R}_\nu^\pi(T) \right] \geq \frac{\sigma_0\sqrt{T}}{32e}$.

This lemma describes that any admissible policy $\pi$ will face some hard instance such that the expected regret could be no less than $\Omega(\sqrt{T})$. Similar as before, we can bridge our problem with the tradition stochastic bandit problem by noticing that $\mathcal{E}_0 \subset \mathcal{E}_1$. It is widely recognized that the stochastic bandit problem has a well-established minimax regret lower bound of $\Omega(\sqrt{T})$ [55]. Intuitively, given that the traditional stochastic bandit problem is a special case within our framework, this lower bound naturally extends to our setting. Formally, by Theorem 2, we have the following theorem, which states that the $\sqrt{T}$ lower bound is indeed possible to achieve and is optimal.

**Theorem 4 (Optimal Regret Rate Design)** *For any $0 < \alpha \leq \frac{1}{4}$, L-EOFU($\alpha$) can guarantee a regret of $\widetilde{\mathcal{O}}(\sqrt{T})$, which can not be further improved up to logarithmic factors. Furthermore, $\inf_\pi \sup_{\nu \in \mathcal{E}_1} \mathbb{E} \left[ \mathcal{R}_\nu^\pi(T) \right] = \widetilde{\Theta}(\sqrt{T})$.*

Theorem 4 reveals that any $\alpha \in (0, \frac{1}{4}]$ yields a homogeneous regret rate dependence on $T$. However, the estimation error, as shown in Theorem 1, decreases with increasing $\alpha$. Consequently, increasing $\alpha$ from nearly 0 to $\frac{1}{4}$ does not significantly increase the regret rate, which remains at $\widetilde{\mathcal{O}}(\sqrt{T})$. However, it significantly reduces the estimation error bound. Therefore, there is limited motivation for decision makers to select $\alpha < \frac{1}{4}$. Nevertheless, there is no free lunch. Even if we choose $\alpha = \frac{1}{4}$, the estimation error can only be bounded by $\mathcal{O}(T^{-\frac{1}{8}})$, which is much larger than the optimal bound of $\frac{1}{\sqrt{T}}$. While the optimal regret rate in this case aligns with that of the traditional stochastic bandit, the optimal design for linear trends faces additional challenges. Intuitively, a regret rate optimal design for linear trends needs to improve from a naive design's $T^2$ regret to the $\sqrt{T}$ order. In contrast, the regret rate optimal design for traditional multi-armed bandits only needs to improve from $T$ regret to $\sqrt{T}$.

From the two cases discussed above, we have uncovered a fundamental trade-off between regret and the statistical power. The design that achieves the optimal regret rate suffers from a large estimation error, while the design that achieves the optimal error rate incurs a large regret. To address this trade-off, we now employ the concept of Pareto optimality from multi-objective optimization theory. This concept allows us to characterize the circumstances where neither regret nor estimation error of the treatment effect can be improved without worsening the other. In other words, we aim to statistically determine the type of optimality that can be achieved between these two extremes.

Formally, we first define the policy class $\Pi_\beta$ as $\Pi_\beta := \{\pi : \exists c_0 > 0, \forall \nu \in \mathcal{E}_1 \text{ and } T, \mathbb{E}[\mathcal{R}_\nu^\pi(T)] \leq c_0 T^\beta\}$, where $c_0$ in this context can depend on universal constants or logarithmic terms of $T$, but it cannot include any polynomial terms of $T$. The set $\Pi_\beta$ encompasses all policies that can achieve a regret of $\widetilde{\mathcal{O}}(T^\beta)$. It is evident that L-EOFU($\alpha$) belongs to $\Pi_{\frac{1}{2} \vee (2\alpha)}$. Our aim is to investigate the best achievable estimation error when the decision maker incurs a regret of at most $\widetilde{\mathcal{O}}(T^\beta)$. The following lemma reveals the inherent difficulty of conducting inference under limited regret budget.

**Lemma 3** *For any $\pi \in \Pi_\beta$, there exists a constant $c^\pi$ only dependent of numerical constants and logarithm terms of $T$, such that for any $\xi > 0$, $\inf_{\hat{\theta}} \max_{\nu \in \mathcal{E}_1} \mathbb{P}_\nu^\pi \left( \left\| \hat{\theta} - \theta_\nu \right\|_1 \geq c^\pi \sigma_0 \xi T^{-\frac{\beta}{4}} \right) \geq \frac{1}{2} - \xi$.*
*Moreover, $\inf_{\hat{\theta}} \max_{\nu \in \mathcal{E}_1} \mathbb{E}_\nu^\pi \left[ \left\| \hat{\theta} - \theta_\nu \right\|_1 \right] \geq \frac{c^\pi \sigma_0}{16} T^{-\frac{\beta}{4}}$.*

Lemma 3 demonstrates that when employing an assignment policy $\pi$ with a regret of $\widetilde{\mathcal{O}}(T^\beta)$, any estimator, regardless of its sophistication, will encounter challenging instances where the estimation error is at least $\widetilde{\Omega}(T^{-\frac{\beta}{4}})$. This finding distinguishes our study, which incorporates linear trends, from the work of [74] that investigates multi-armed bandit experimental design without linear trends. In the absence of linear trends, [74] reveals that under an assignment policy $\pi$ with a regret of $\widetilde{\mathcal{O}}(T^\beta)$, the estimation error has a minimax lower bound of $\widetilde{\Omega}(T^{-\frac{\beta}{2}})$. The key distinction arises from the fact that linear trends can significantly increase the cost of exploration, resulting in higher regret during the exploration phase. Consequently, with the same regret budget, the information collected in our case may be substantially less than in the absence of linear trends. The proof follows this intuition by constructing two similar instances with the same optimal arm but slightly different suboptimal arms. The only way to obtain useful information to distinguish the two is by playing the suboptimal arm. Given a regret budget of $T^\beta$, at most approximately $T^{\frac{\beta}{2}}$ samples can be obtained from the suboptimal arm, leading to a lower bound of $T^{-\frac{\beta}{4}}$.

**Theorem 5 (Pareto Optimal Design)** *If $\alpha \geq \frac{1}{4}$, `L-EOFU`$(\alpha) \in \Pi_{2\alpha}$, and thus, the $T^{-\frac{\alpha}{2}}$ error rate in Theorem 1 can not be further improved in terms of the dependence on $T$. Furthermore, for a fixed $\beta \geq \frac{1}{2}$, $\inf_{\hat{\theta}_T, \pi \in \Pi_\beta} \max_{\nu \in \mathcal{E}_1} \mathbb{E}_\nu^\pi \left[ \left\| \hat{\theta}_T - \theta_\nu \right\|_1 \right] = \widetilde{\Theta}\left( T^{-\frac{\beta}{4}} \right)$.*

The theorem establishes that the regret bound of $T^{-\frac{\alpha}{2}}$ achieved by `L-EOFU`$(\alpha)$ cannot be improved by a constant factor, given that the decision-making policy belongs to $\Pi_{2\alpha}$. This implies that `L-EOFU`$(\alpha)$ is Pareto optimal for any $\alpha \geq \frac{1}{4}$. The reasons are as follows: on the one hand, if one desires a better guarantee on the estimation error than $T^{-\frac{\alpha}{2}}$, the only option is to accept a higher regret rate by adopting a larger decision-making policy class $\Pi_{2\alpha+\epsilon}$ for some $\epsilon > 0$. On the other hand, if a regret of $\widetilde{\mathcal{O}}(T^{2\alpha-\epsilon})$ is aimed for, the estimation error will inevitably increase to the order of $T^{-2\alpha+\epsilon}$. In other words, improving either the statistical power of causal inference or the efficiency of online decision-making necessitates a degradation in the other. Intuitively, Theorem 5 captures a statistical Pareto frontier for the trade-off between conducting causal inference and minimizing regret, which can be expressed as Estimation Error $= \Theta(\text{Regret}^{-\frac{1}{4}})$. In Theorem 3, we reveal that $\alpha = 1$ leads to the optimal design in terms of error rate. By combining Theorem 5, we further conclude that the large $T^2$ regret incurred by $\alpha = 1$ is necessary for any estimator achieving the optimal error rate of $\frac{1}{\sqrt{T}}$. Moreover, the $T^{-\frac{1}{8}}$ error rate achieved by $\alpha = \frac{1}{4}$, although relatively large, is the best that any regret-optimal policy can achieve. Figure 3 illustrates the Pareto frontier under linear trends, alongside the Pareto frontier for standard multi-armed bandit experimental design studied in [74].

From Figure 3, we can observe the significant impact of the non-stationary structure on optimal experimental design. First, in standard bandit experiments, the maximum possible regret is of the order $T$ since all parameters are assumed to be bounded. However, with the presence of linear trends, traditional designs such as switchback experiments are likely to incur regret of the order $T^2$. Furthermore, under the same regret budget, the best achievable estimation error bounds differ substantially. For instance, given assignment policies with regret of the order $T$, the

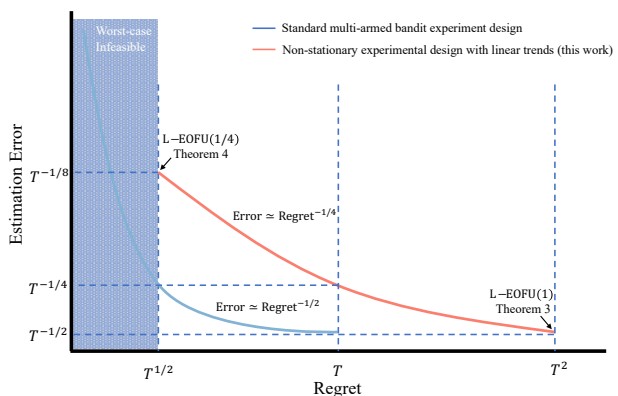

Figure 3: The Pareto Frontier.

optimal design for standard bandit experiments can attain an error bound of $T^{-1/2}$, whereas in the presence of linear trends, the best achievable bound is $T^{-1/4}$. Intuitively, this is because linear trends make gathering information from the suboptimal arm increasingly costly in terms of regret. Similarly,

when linear trends exist, the decision maker will always have to pay additional regret to obtain an estimator with the same level of precision. Additionally, there is a worst-case infeasible region as indicated by Lemma 2 that no policy can achieve a regret smaller than $\sqrt{T}$ order in the worst case.

## 3  Discussion

In this section, we'll discuss the limitations and future directions of our work. Firstly, the assumption of prior knowledge about non-stationarity following a linear form is strong. While it provides a starting point for understanding experiment design under structured non-stationarity, practical approaches are needed. One option is to use historical data for prior information, if there exists some. Alternatively, we can use data collected during the first phase of our L-OFUE algorithms to verify trends. Another question is model misspecification. What if the true underlying model is non-parametric, but the decision maker prefers using interpretable linear functions? Understanding how our design performs in such scenarios will provide valuable insights into its practical applicability. Furthermore, our framework and results are formulated based on a specific notion of non-stationarity. Exploring whether our framework can be adapted to different formulations of non-stationarity is another area of interest for future research. Another important direction for our future research is the exploration of instance-dependent regret bounds under linear and polynomial trends. In this paper, we focus on studying the worst-case optimal regret. However, in the field of classical stochastic bandits, there has been increasing attention on instance-dependent regret bounds (e.g., [11] [33][5]). It would be valuable to extend our analysis to incorporate instance-dependent parameters. Defining suitable instance-dependent parameters is not a trivial task, as the classical parameters such as arm gaps may not be directly applicable in our setting, where the parameters themselves change over time. Another more practical question that people may be of interest is how to choose $\alpha$. One possible approach is to adaptively end the first phase of the experiment. Ideally, continuous monitoring during Phase 1 would allow for the experiment to halt as soon as certain criteria are met, eliminating the need to manually select $\alpha$. However, reliable and efficient continuous monitoring of the switchback experiment poses significant challenges. Even in basic A/B testing without a linear trend, designing continuous monitoring requires substantial effort and delicate power analysis, as demonstrated in [45].

## 4  Conclusion

In this work, we present a comprehensive investigation into the problem of non-stationary experimental design with linear trends. Our focus is on two objectives: accurate estimation of dynamic treatment effects and minimizing welfare loss within the experiment. For scenarios involving linear trends, we propose an efficient design called L-EOFU. This design can be customized to achieve optimal estimation error rates, optimal regret rates, or strike a Pareto optimal trade-off between the two objectives. To establish the efficacy of our approach, we provide information-theoretical lower bounds that highlight the inherent challenges in estimating dynamic treatment effects and minimizing welfare loss. Additionally, we unveil the fundamental trade-off between the two objectives.

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
