# OpenReview forum: "Non-stationary Experimental Design under Linear Trends"
_NeurIPS.cc/2023/Conference — NeurIPS 2023 poster_

### Official Review · Reviewer_sYAZ · 2023-07-04

**Soundness:** 3 good
**Presentation:** 4 excellent
**Contribution:** 2 fair
**Rating:** 6
**Confidence:** 4

**Summary:**

The paper introduces a new experimental design for non-stationary experiments under the assumption that the expected potential outcomes are linear in time.

The proposed design consists of two phases: a switchback experiment that alternates at each time point and a UCB-type algorithm for multi-armed bandits. Phase 2 incorporates information learned from Phase 1, but the inference is solely based on Phase 1 data.

The authors also discuss the following theoretical properties of the experiment:
 - an upper bound on estimation error,
 - an upper bound on regret,
 - an optimal design that demonstrates Pareto optimality within the considered design class.

**Strengths:**

The paper explores a scenario that hasn't been studied before, and the proposed design is also novel.

As far as I know, the regret bound in the paper has not been derived before under this specific set of assumptions.

The observation that there is not a strict tradeoff for choosing $\alpha$ in $(0,1/4]$ is interesting and insightful.

The paper is well-written and easily understandable.

The design and inference presented in the paper are simple and practical.

**Weaknesses:**

The assumption of a linear trend is not compelling enough to me:
 - The authors have not provided any realistic example under which such a linear trend might occur
 - The authors mentioned that a drawback of current works on decision-making under non-stationarity is the lack of knowledge transferability to future, but it is not clear to me whether the treatment effect can really be extrapolated through a linear function.

The class of designs considered in the paper appears to be quite specific. It might benefit from additional motivation for this particular set of designs:
 - Why use a switchback for the first phase? Under the given assumption, it seems that it doesn’t matter when we assign treatment and control, as long as there are $T^\alpha/2$ periods assigned to each group. While a switchback design may offer robustness in certain scenarios where the assumption might not hold, the authors have not adequately explained this aspect in the current version of the paper (Please also refer to the comments below for robustness concerns).
 - One important contribution of the paper is the observation in Theorem 5 regarding the Pareto optimal design, but it is only optimal contingent on the specific class of designs and estimation strategy considered. For instance, although post-bandit inference is challenging, there is certainly information in the data that can be utilized and there are works on it. Would the same conclusion still hold if the inference were not solely based on data collected in Phase 1?
 - Under which circumstances would the practitioners find such an experiment to be of interest? Why couldn’t they simply keep running a bandit algorithm? If they want to do inference, it is still possible to do post-bandit inference, or they may just pause the algorithm and run a randomized experiment.

Robustness concerns:
 - It seems that the inference could be misleading if the linear assumption is off. This is especially concerning since the insight into the post-experiment period is solely based on information from the first part of the experiment, which can be a very strong statement under nonstationarity.
 - How would a practitioner know if the linearity assumption is reasonable? What if, during Phase 1, they observe that there is actually not such a linear trend?
 - Related to the motivation for the switchback, what is the reason for implementing a one-coin flip (i.e., having a deterministic sequence of actions) instead of opting for a completely randomized experiment (which is in general more robust in an adversarial environment)?

**Questions:**

See weaknesses.

**Limitations:**

See weaknesses.

---

> ### Author Rebuttal · Authors · 2023-08-09
>
> Thank you for your valuable feedbacks! We refer our response to the linear assumptions, the extension to higher order polynomial trends, and the switchback design to the "global response" on top of page. We detail responses to the other comments as follows.
>
> a.	About the Pareto optimality.
>
> Sorry that we may have not fully digested the comment “it is only optimal contingent on the specific class of designs and estimation strategy considered.” In the follows, we are responding according to our understanding. Kindly correct us if we have misinterpreted the comment.
>
> First, we may want to repeat ourselves a bit that when we refer to a design as Pareto optimal, it signifies that there exists no other design that can strictly outperform it on both objectives (i.e., regret and estimation error). Pareto optimality is an intrinsic property of a design and is not confined to a specific class of designs and estimation strategies. It necessitates comparing all potential sophisticated designs to determine their optimality. Our design represents a series of Pareto optimal designs. One of the main technical challenges we faced was ensuring that our design cannot be strictly outperformed by any other possible policy.
>
> Second, with regard to the post-bandit inference, there may be valuable information in the data that can be utilized. However, our lower bound in Lemma 3 indicates that the information we "waste" does not fundamentally alter the essential outcome. Specifically, Lemma 3 establishes a lower bound for all admissible estimators by taking the infimum over all possible $\hat{\theta}_T$. This means that for any estimators, regardless of whether they utilize all the data (including those based on post-bandit inference), the estimation errors are lower bounded by a constant in the order of $T^{-\frac{\beta}{4}}$, given that the data are generated from an assignment policy with regret bounded by $T^\beta$. In essence, the unused information may marginally reduce the estimation error concerning the constant term, but not the $T$ dependence. For instance, the estimation error may decrease from $T^{-\frac{\beta}{4}}$ to $0.8T^{-\frac{\beta}{4}}$.  Exploring ways to further reduce the constant by utilizing data collected in Phase 2 is a crucial aspect of our future work. As you astutely pointed out, addressing this task is likely to be quite challenging, as evidenced by the extensive work conducted by pioneers Zhang et al. [D] in solving the task for the batched bandit. Hence, we humbly request to defer this aspect to our future work.
>
> We should have used $\hat{\theta}$ instead of $\hat{\theta}_T$ in both Lemma 3 and Theorem 5. We hope that this inconsistency did not cause any confusion in understanding.
>
> b. Alternatives to our current design.
>
> We appreciate your proposal of a post-bandit inference and the idea of pausing a bandit algorithm to run a randomized experiment. Both are indeed inspiring and worthy of consideration. Meanwhile, we like to highlight several essential reasons that deterred us from pursuing these alternative routes.
>
> We have explained a bit in our response c above that post-bandit inference may not make a huge difference. Specifically, if we run the rate optimal bandit algorithm, we will obtain $T^{1/4}$ samples from the suboptimal arm. Even employing several different techniques including bootstrap or others, it is unreasonable to expect that the estimators based on $T^{1/4}$ samples could enjoy the same inference quality as those based on $T$ samples. Additionally, the assumptions commonly made in post-bandit inference literature, such as lower-bounded probabilities of arm selection, may not hold when we adopt OFU-type algorithms. Consequently, traditional post-bandit inference techniques may not be easily applicable to our setting.
>
> As for pausing a bandit algorithm to run a randomized experiment, defining the stopping time presents a significant challenge. Continuous monitoring throughout the entire experiment is required to determine when to pause the algorithm, making the process complex. Achieving continuous monitoring is already difficult in basic A/B testing without adaptivity, as highlighted in a recent work [G]. It would be even more challenging in our setting. Furthermore, the total regret incurred by the proposed method with the randomized experiment lasting for $T^{\alpha}$ periods may become very large. For instance, when the expected outcomes for the two arms at time $t$ are $\mathbb{E}[r_t(1)]=t$ and $\mathbb{E}[r_t(2)]=2t$, the regret incurred during the randomized experiment would be in the order of $T^{1+\alpha}$. In contrast, our design incurs a regret of only $T^{2\alpha}$ during the exploration phase, which is always smaller since $\alpha \leq 1$. Thus, obtaining the same amount of information would result in different incurred regrets. Additionally, designing a bandit algorithm under the linear trend poses its own set of challenges. Our analysis heavily relies on the data collected from the first phase. Since the rewards cannot be bounded by a constant independent of $T$, which we have emphasized in the paper, many traditional algorithms and techniques may not work, and this has been a significant technical challenge for us to address.
>
> In conclusion, while we acknowledge the appeal of the alternative ideas that you proposed, we have thoroughly considered the inherent difficulties and limitations associated with them. We will ensure to provide a more comprehensive discussion of these alternative ideas, along with a thorough explanation of why our current approach was chosen.
>
> References:
>
> [D] Zhang et al. (2020). Inference for batched bandits. NeurIPS 2020.
>
> [E] Bibaut et al. (2021). Post-contextual-bandit inference. NeurIPS 2021.
>
> [F] Hadad et al. (2021). Confidence intervals for policy evaluation in adaptive experiments. PNAS.
>
> [G] Johari et al. (2022). Always valid inference: Continuous monitoring of a/b tests. Operations Research, 70(3), 1806-1821.

---

> > ### Comment · Reviewer_sYAZ · 2023-08-19
> >
> > Thank you for the clarifications. I take back my comments on the Pareto optimal design. The class of estimators considered is indeed more general than I thought. Although post-selection inference can improve the rate of reward estimation associated with the optimal arm, it won't help much in terms of treatment effect estimation. I have raised my scores accordingly.
> >
> > The theoretical results on generalization to polynomial trends are interesting, but I don't think higher-order polynomial trends are in general practical. Taylor expansion provides only a local approximation, so it can be very unstable for approximating effects in the long term.

---

> > > ### Author Response · Authors · 2023-08-20
> > >
> > > We really appreciate your time and reading our response. It is our great pleasure to know that the response clarifies your questions and comments.
> > >
> > > Regarding the generalizing polynomial trends that we proposed in the response as a potential extension, we are in complete agreement with your assessment. Utilizing Taylor expansion for approximating long-term effects may indeed introduce instability for the long term. We acknowledge that the non-stationary experiment design area needs more general extensions, and we plan to focus on them in the next phase of our research. We hope that our current model, though not a perfect one, could be viewed as a first step toward more complicated non-stationary trends and may provide some useful technical tools.
> > >
> > > We greatly value your feedback, and it's a big help in making our work better.

---

### Official Review · Reviewer_GLy4 · 2023-07-07

**Soundness:** 4 excellent
**Presentation:** 3 good
**Contribution:** 3 good
**Rating:** 6
**Confidence:** 4

**Summary:**

This work is concerned with designing an adaptive experiment in the presence of a linear or polynomial trend in the potential outcome. The authors describe the natural tension between estimation estimation error and regret and describe the Pareto optimal objective. They then introduce an optimistic procedure that seeks to optimize for each of the three objectives depending on the value of a free parameter. It is then shown that the procedure achieves the optimal error rate and optimal regret. Both of which are dependent on the choice of the free parameter $\alpha$. Finally, the authors show that the proposed procedure is Pareto optimal.

**Strengths:**

* This is a well motivated, and to my eyes, novel task.
* The authors do a nice job of both describing the problem and clearly laying out the inherent difficulty in addressing the problem (the tradeoff between estimation error and regret).
* The proposed algorithm is intuitive, and the properties are appealing.
* The technical results are well written, and should be fairly accessible to readers.
* I can envision this work having impact on practice, where it is common for practitioners to encounter the phenomenon of potential outcomes that change over time.

**Weaknesses:**

The largest concerns here are:
1. The authors require the selection of $\alpha$ for all of the analyses. This is reasonable, given the tradeoff. However, in practice it would seem that it is going to be difficult for experimenters to choose this a priori.
2. The authors assume a well-specified model, i.e. the underlying trend is truly linear or polynomial. This seems to my eyes to be very simplified. While likely out of scope of the paper to provide robustness properties of the proposed procedure, it would seem important to have a discussion of this within the text.
3. Given (1) and (2), it would have been nice to see some kind of empirical evaluation that shows both the performance of the algorithm against naive alternatives and the relative robustness
4. (minor) The author references in supplement C are broken.

**Questions:**

* Can the authors give a sense of guidance for setting $\alpha$ in practice?
* Is it possible to provide a set of simple simulation results that show the empirical behavior of the proposed algorithm?


**Limitations:**

Yes, the authors do a very nice job of describing the limitations.

---

> ### Author Rebuttal · Authors · 2023-08-09
>
> Thank you for your valuable feedback and helpful comments on our work. We sincerely appreciate your careful read of our paper, and we are grateful for the opportunity to address your questions and concerns. Below, we provide our detailed response:
>
> 1.	The Selection of $\alpha$.
>
> We agree with your suggestion, and we recognize the importance of discussing the selection of $\alpha$ more thoroughly. While choosing the optimal value of $\alpha$ may be a challenging task, we acknowledge several practical and intuitive approaches that could be employed.
>
> One option is for the experimenter to determine the largest tolerable estimation error, for instance, in the order of 0.1, and with $T$ experimental units available, select $\alpha$ as $2\log_{T}10$. This approach leverages prior information on the acceptable level of estimation error, providing a basis for the selection of $\alpha$. Similarly, if the primary objective is to strictly control the loss of welfare, the choice of $\alpha$ can be made accordingly. These practical approaches, while not without their limitations, offer feasible methods to decide on an appropriate value for $\alpha$, taking into account specific requirements and constraints.
>
> Another approach we considered is to adaptively end the first phase of the experiment. Ideally, continuous monitoring during Phase 1 would allow for the experiment to halt as soon as certain criteria are met, eliminating the need to manually select $\alpha$. However, reliable and efficient continuous monitoring of the switchback experiment poses significant challenges. Even in basic A/B testing without a linear trend, designing continuous monitoring requires substantial effort and delicate power analysis, as demonstrated in [A]. While we recognize the potential benefits of this adaptive approach, we consider it an essential future direction, and we aim to explore this avenue in our future work.
>
> 2.	Generalization to higher order polynomial trends.
>
> We appreciate your positive assessment of the generalization of our L-EOFU algorithm and the associated lower bounds to accommodate higher order polynomial trends. The established result, where the trend follows a $k$th order polynomial, indicating a best estimation error of $T^{-\beta/(2k+2)}$ for any design with a regret bound of $T^\beta$, allows for enhanced flexibility and adaptability. Specifically, Phase 1 can now serve the additional purpose of observing the optimal choice of $k$, providing data-driven insights for decision-making.
>
> Additionally, leveraging Taylor expansion enables us to handle more general functions by choosing a sufficiently large value of $k$. This approach represents a potential solution to address the misspecification issue, and we will incorporate a thorough discussion of the model misspecification within the text.
>
> Regarding the simulation provided in Appendix A, it serves the purpose of illustrating the trade-off between minimizing the estimation error and maximizing statistical power. We will fix the reference issue in Supplement C to ensure the accuracy and reliability of our work. Once again, we extend our gratitude for your valuable time and insightful comments. Your feedback has been instrumental in enhancing the quality and clarity of our paper.
>
> Reference:
>
> [A] Johari, R., Koomen, P., Pekelis, L., & Walsh, D. (2022). Always valid inference: Continuous monitoring of a/b tests. Operations Research, 70(3), 1806-1821.

---

### Official Review · Reviewer_EcQ7 · 2023-07-09

**Soundness:** 3 good
**Presentation:** 2 fair
**Contribution:** 2 fair
**Rating:** 5
**Confidence:** 3

**Summary:**

This paper studied the problem of non-stationary experimental design. The authors adopt a linear model assumption, where the reward associated with each arm exhibits a linear relationship with time. They concern two objectives: accurate estimation of dynamic treatment effects and regret. They propose a two-stage Exploration and OFU design algorithm that can be customized for optimal estimation error rate, optimal regret rate or the Pareto optimal trade-off between the two objectives. The paper also provides information-theoretic lower bounds to illustrate the challenge of achieving two objectives simultaneously. The experimental results on synthetic data demonstrate is consistent with theoretical results.

**Strengths:**

1. The paper is well-written and the overall idea of the paper was clear and easy to understand.

2. The theoretical results on the information-theoretical lower of causal estimation and regret seem technically sound.

**Weaknesses:**

1. The author mentions that [1] also investigates the trade-off between regret and causal inference using Pareto optimality. It seems that [1] already introduce the concept of Pareto optimality to characterize the trade-off between estimation error and regret, which might show the lack of technical novelty in the problem formulation. Therefore, additional elaborations are necessary to elucidate the connection and difference between the present work and [1].

2. Moreover, it is worth noting that [1] does not impose any assumptions on the reward model. In contrast, the current paper introduces a linear structure, which constitutes a rather strong assumption. While I acknowledge that this work extends the analysis to a non-stationary setting, it would be valuable to explore the possibility of devising a more general model class that can accommodate diverse reward specifications. By doing so, the research would not only enhance its applicability but also contribute to the advancement of the field by providing a broader framework for analysis.

3. Additional numerical experiments would greatly enhance the paper‘s empirical evaluation. As it stands, the paper solely encompasses a single scenario, which limits the scope of its findings. To address this concern, I recommend the authors incorporate multiple scenarios in their analysis, thereby demonstrating the robustness of the proposed algorithm across various settings. For instance, investigating the impact of varying $\sigma_a^2$ on the results could provide valuable insights into the algorithm’s performance under different conditions.

4. While the current paper provides a thorough analysis focusing on two arms, it is necessary to discuss the extension of the proposed framework to accommodate settings of more than 2 arms.


[1] Simchi-Levi, David, and Chonghuan Wang. “Multi-armed bandit experimental design: Online decision-making and adaptive inference.” In International Conference on Artificial Intelligence and Statistics, pp. 3086-3097. PMLR, 2023.

**Questions:**

Please consider addressing questions in Weaknesses.

**Limitations:**

Yes

---

> ### Author Rebuttal · Authors · 2023-08-09
>
> We sincerely appreciate your helpful comments and careful read of our work. Thank you for providing us with an opportunity to address your concerns.
>
> 1.	Comparing with [A].
>
> We completely agree that [A] was the first to introduce Pareto optimality to characterize a similar trade-off. Allow us to clarify that we do not claim, nor will we plan to claim, that our contribution includes the introduction of Pareto optimality. Instead, we like to highlight the technical novelty and several significant points that differentiate our work as follows. The Pareto trade-off, serves as a metric, instead of a result, to illustrate our work’s contribution.
>
> Firstly, in terms of formulation, our main contribution lies in capturing non-stationarity with linear trends, which, to the best of our knowledge, is a novel concept in the related literature on experimental design. It is essential to note that [A] assumes stationarity of the outcomes, and when the linear trend is zero, the outcome structure we consider encompasses the structure examined by [A]. Therefore, from a formulation perspective, [A] can be seen as a special case of our work, which did not consider non-stationarity.
>
> Secondly, in the algorithm design, our problem formulation is new both in the experimental design literature and within the bandit literature. As a consequence, traditional algorithms and proof techniques for bandit problems may not be directly applicable. As discussed in the paper, the main technical challenges stem from the fact that outcomes can no longer be assumed to be bounded by a universal constant (independent of time horizon $T$), as is common in much of the bandit literature. This unboundedness necessitates the development of novel techniques to ensure the performance of the regret bound of our algorithm. To be more precise, the unboundedness issue renders the regret of several traditional MAB algorithms to be in the order of $T$ and fall short of achieving the optimal regret rate of $\sqrt{T}$ (see lines 261 to 270 in the paper). In contrast, [A] studies the fundamental MAB experiment case, where traditional MAB algorithms and analysis can be applied.
>
> Thirdly, in terms of results, Figure 3 in the paper illustrates the differences between our work and [A]. While [A] establishes that the trade-off can be approximately captured by $error \simeq regret^{-1/2}$, our findings reveal that with a linear trend, the trade-off becomes $error \simeq regret^{-1/4}$. This means that given the same regret budget, estimating the model with a linear trend is more challenging. Additionally, due to the unbounded issue, the regret incurred by a naive algorithm could even be in the order of $T^2$, as opposed to the order of $T$ in [A]. Further elaboration on the distinctions between [A] and our work can be found in the paper from lines 391 to 411.
>
> In the revised version, we plan to emphasize these points more explicitly to distinguish ourselves from [A]. We sincerely hope that this response alleviates potential concerns you may have had.
>
> 2.	Generalization to higher order polynomial trends.
>
> Our L-EOFU algorithm and the associated lower bounds can indeed be extended to accommodate cases where the trend is formulated by a higher order polynomial function. Specifically, when the trend follows a $k$th order polynomial, we can establish a general result that provides the best estimation error one can expect for any design with a regret bound of $T^\beta$ to be $T^{-\beta/(2k+2)}$. It is noteworthy that when $k=1$, this result aligns precisely with what we presented in the paper. Therefore, the generalized formulation covers and extends the findings in the submitted work. Moreover, during the Phase 1 in our design, it is possible to observe and determine the optimal choice of $k$ for a given setting. This allows for a data-driven approach to decide which polynomial order best fits the underlying trend.
>
> To address more complex functions beyond polynomial trends, we can leverage Taylor expansion. By selecting a sufficiently large value of $k$, we can effectively approximate and handle a broader range of functions, thereby increasing the versatility of our approach.
>
> In conclusion, we are excited about the potential of solving higher order polynomial trends under our framework, as it presents an avenue to expand the scope of applications.
>
> 3.	Generalization to $K$ arms.
>
> We agree that extending our approach to settings with more than 2 arms, such as $K$ arms, would be beneficial. Our current framework allows generalization to $K$ arms. Allow us to outline the necessary modifications and improvements for this generalization. In the context of more than 2 arms, we introduce the gap between arm $i$ and arm $j$ as $\theta_\nu^{(i,j)}$. To address the estimation error, we simply replace the original $\left|\hat{\theta}-\theta_\nu\right|$ with $\max_{i<j\le K} \left|\hat{\theta}^{(i,j)}-\theta_\nu^{(i,j)}\right|$, which considers the maximum deviation between the estimated and true values across all pairs of arms. Fortunately, all the theoretical results follow a similar structure, allowing for a seamless adaptation to this extended setting.
>
> Furthermore, the L-EOFU design showcases flexibility in coping with more than 2 arms. For Phase 1, the algorithm efficiently explores all $K$ arms sequentially, ensuring a comprehensive initial exploration. In Phase 2, the approach remains largely unchanged, with the only difference being the need for estimations for all $K$ arms. As a result, the generalization to multiple arms does not impose additional technical challenges.
>
> We agree with your suggestion to incorporate additional numerical experiments in the revised version. We genuinely appreciate your insightful comments. We will diligently incorporate these enhancements and extensions into the revised version.
>
> Reference:
>
> [A] Simchi-Levi D, Wang C. “Multi-armed bandit experimental design: Online decision-making and adaptive inference.” AISTATS 2023.

---

### Author Rebuttal · Authors · 2023-08-09

We thank all the reviewers for your time, valuable feedback, and insightful comments. In addition to the responses to each reviewer, we would like to use this part to provide clarifications on three specific comments.

a.	Assumption on linear trends.

Linear trends are indeed a strong assumption, and they may not always fully represent the complexities of real-world treatment effects. That said, we consider linear trends to be a crucial starting point in understanding structured non-stationary trends when conducting experiments. As the first work to explore non-stationary trends in experiments, upon the standard practice of viewing experiments as stationary, we think that focusing on linear trends offers a fundamental perspective to understand how non-stationary trends affect experiments in various aspects.

While linear trends might not be the most sophisticated model, they have proven to be highly useful for understanding and predicting certain phenomena across various fields. Allow us to provide several illustrative examples to support this claim:

1.	(Epidemiology) Wang et al. [A] observed a linear trend in the prevalence of obesity and overweight among US adults, based on national survey data collected between 1970s and 2004. This linear trend served as a benchmark for predicting the prevalence from 2004 to 2050, influencing numerous epidemiology studies.
2.	(Ecology) [B] conducted a study on monitoring and predicting the California sea otter population using linear trends observed in aerial surveys, showcasing the usefulness of linear trends in ecological studies.
3.	(Atmospheres) Nilsson and Elgered [C] revealed a linear trend in the integrated water vapor content based on a 10-year ground-based GPS data, highlighting the application of linear trends in atmospheric research.

In these examples, linear trends are not only used to understand immediate changes but also to predict and assess the long-term implications of various policies and interventions. While we acknowledge that linear trends may not capture all nuances, they do provide valuable insights, especially when prior information suggests approximate linearity. Linear trends, admittedly, are not able to capture all kinds of non-stationary trends, but can serve as a first step toward more complicated non-stationary trends. We hope that our work's consideration on linear non-stationary trends complements the previous literature on experimental design with no non-stationarity, and builds some useful tools for the consideration of other non-stationary trends.

We understand the importance of providing realistic examples to support our claims, and we apologize for the oversight in not including specific real-world scenarios in our initial submission. In the revised manuscript, we will diligently address this concern by incorporating the mentioned scenarios where linear trends play a significant role. Additionally, we will emphasize the applicability of our framework in situations where linear models are preferred due to their strong interpretability.

b.	Generalization to higher order polynomial trends.

We agree with your suggestion, and we are delighted to confirm that our L-EOFU algorithm and the associated lower bounds can indeed be generalized to accommodate higher order polynomial trends. Specifically, when the trend follows a $k$th order polynomial, we have established a general result that provides the best estimation error one can expect for any design with a regret bound of $T^\beta$ to be $T^{-\beta/(2k+2)}$.

It is important to note that when $k=1$, this result aligns precisely with what we presented in the main paper. Thus, the generalized formulation encompasses and extends the findings we reported earlier. Given this generalization, Phase 1 can be effectively utilized to observe and determine the optimal choice of $k$. Through this approach, the experimenter can assess which polynomial order best fits the underlying trend and make informed decisions accordingly. This flexibility allows for a data-driven selection of the appropriate polynomial order, which can significantly alleviate concerns about the possibility of observing a non-linear trend during Phase 1.

Furthermore, we acknowledge the significance of Taylor expansion in dealing with more complex functions beyond simple polynomial trends. By choosing a large enough value of $k$, the Taylor expansion enables us to effectively approximate and handle a broader range of functions, thereby enhancing the applicability of our approach.

In conclusion, we are thrilled about the generalization to higher order polynomial trends. It not only addresses your concern about the linearity assumption but also allows for a data-driven and flexible selection of the polynomial order.

c. The switchback design in the first phase.

We appreciate that some reviewer points out as long as predetermined $T^\alpha/2$ periods are assigned to each group, our design would be effective. We acknowledge that the switchback design is just one of the possible approaches and, indeed, we opted for it primarily to facilitate the theoretical analysis. Specifically, in the proof to Theorem 1 in Appendix B, the switchback design offers advantages by enabling us to explicitly derive expressions for $V_{1,0}$, $V_{2,0}$, and their inverses. This explicit representation brings significant convenience to the analysis and helps in drawing insightful conclusions. We acknowledge that other designs could be equally valid in practice.

References:

[A] Wang et al. (2008). Will all Americans become overweight or obese? Estimating the progression and cost of the US obesity epidemic. Obesity, 16(10), 2323-2330.

[B] Gerrodette, T. I. M. (1987). A power analysis for detecting trends. Ecology, 68(5), 1364-1372.

[C] Nilsson, T., & Elgered, G. (2008). Long‐term trends in the atmospheric water vapor content estimated from ground‐based GPS data. Journal of Geophysical Research: Atmospheres, 113(D19).

---

### Decision · Program_Chairs · 2023-09-21

**Decision:**

Accept (poster)

**Comment:**

The paper studies an adaptive experimental design to handle the case of non-stationary outcomes but subject to a linear trend. Both the problem and the approach are new and nicely formulated/executed, as recognized by the reviewers. The authors should implement their responses into the text in preparing a camera ready. For example, additional literature on inference after adaptive experimentation and always valid inference was brought up during discussion and should be discussed in the paper.